# Measurements and Modelling of Thermally Induced Warpages of DIMM Socket Server PCB Assembly after Solder Reflow Processes

**DOI:** 10.3390/ma16083233

**Published:** 2023-04-19

**Authors:** Ming-Yi Tsai, Yu-Wen Wang, Yen-Jui Lu, Tzu-Min Lu, Shu-Tan Chung

**Affiliations:** 1Department of Mechanical Engineering, Chang Gung University, Taoyuan 33302, Taiwan; wanglovedog@gmail.com (Y.-W.W.); randy12226@gmail.com (Y.-J.L.); 2Inventec Corporation, Taoyuan 333547, Taiwan; liu.lawrence@inventec.com (T.-M.L.); chung.kelvin@inventec.com (S.-T.C.)

**Keywords:** thermal warpage, DIMM socket, printed circuit board, strain gauge, shadow moiré, finite element method

## Abstract

The thermal warpage of a server-computer-used DIMM socket-PCB assembly after the solder reflow process is studied experimentally, theoretically, and numerically, especially along the socket lines and over the entire assembly. Strain gauge and shadow moiré are used for determining the coefficients of thermal expansion of the PCB and DIMM sockets and for measuring the thermal warpages of the socket-PCB assembly, respectively, while a newly proposed theory and a finite element method (FEM) simulation are used to calculate the thermal warpage of the socket-PCB assembly in order to understand its thermo-mechanical behavior and then further identify some important parameters. The results show that the theoretical solution validated by the FEM simulation provides the mechanics with the critical parameters. In addition, the cylindrical-like thermal deformation and warpage, measured by the moiré experiment, are also consistent with the theory and FEM simulation. Moreover, the results of the thermal warpage of the socket-PCB assembly from the strain gauge suggest a warpage dependence on the cooling rate during the solder reflow process, due to the nature of the creep behavior in the solder material. Finally, the thermal warpages of the socket-PCB assemblies after the solder reflow processes are provided through a validated FEM simulation for future designs and verification.

## 1. Introduction

The thermally induced warpages of integrated-circuit (IC) package- or component-PCB (printed circuit board) assemblies after the solder reflow processes, due to the mismatch of the coefficients of thermal expansion (CTEs) of those packages or components with the PCB, create problems in the later-on assembly and the solder joint reliability under in-service thermal cycling [1]. Therefore, measurement and analysis approaches to this problem are of importance and needed in order to provide further control and reduction in the thermal warpage of the package- or component-PCB assembly after the reflow processes. Reliable and easy-to-use strain gauges [2] have been used in measuring the mechanical strains on PCBs and packages [3] and on the CPU assembly [4], as well as the thermally induced curvatures and warpages of the PCB [5], flip-chip, and 2.5 D IC packages [6]. Additionally, a shadow moiré method [7,8,9] for measuring the full-field out-of-plane deformations of the specimens has been also used for the PBGA packages under thermal loading [10,11,12]. Both well-developed experimental methods will be applied in this study as well. The configuration and detailed geometry of a dual in-line memory module (DIMM) socket-PCB assembly used in a server computer system is shown for this study in Figure 1, including a socket for the DIMM in Figure 1a with 288 Cu pins, a socket-PCB assembly with 24 DIMM sockets in Figure 1b representing the full-assembly specimen of interest, and a solder joint of a Cu pin for the DIMM socket on the PCB in Figure 1c. Excessive thermal warpage of the DIMM socket-PCB assembly after the solder reflow processes could generate assembly and reliability problems. The problems include the difficulty of a latch engagement and the shift of contact points with gold fingers in the DIMM socket, and a later-on installation with a flat metal plate on the full socket-PCB assembly during structural reinforcement. Therefore, the thermal warpage of the DIMM socket-PCB assembly along the socket lines and over the entire assembly after the solder reflow process will be thoroughly investigated experimentally, theoretically, and numerically in this study.

## 2. Methodologies

The methods used in this study include experimental measurements and theoretical and finite element analyses. In the experimental measurements, strain gauge and shadow moiré were used for determining the CTEs of the PCB and DIMM socket as well as for measuring thermal warpages of the socket-PCB assembly, respectively, while theoretical and finite element analyses were used to calculate the thermal warpage of the socket-PCB assembly and further identify important parameters in order to understand its thermo-mechanical behavior. These methods will be briefly illustrated in this section.

### 2.1. CTE and Thermal Deformation Measurement by Stain Gauges

The typical strain gauge measurement [2] at elevated temperatures can be given as
(1)$$\varepsilon_{a}=\varepsilon_{t}+\frac{\left(\gamma_{g}+\gamma_{w}\right)}{S_{g}}\Delta T$$
where ε*_a_* is an apparent strain which is directly obtained from a strain gauge measurement system, ε*_t_* is a true strain which is an actual strain on the measured point of the specimen, *S_g_* is a gauge factor, Δ*T* is thermal loading, and *γ_g_* and *γ_w_* are the temperature coefficients of resistivity of gauge grid metal material and connected lead wire, respectively. 

For measuring the thermal strains of the PCB during the heating and cooling processes, back-to-back strain gauges were adhered at a specific position on both the top and bottom surfaces of the PCB. The apparent strains on the top and bottom surfaces (ε*_a,top_* and ε*_a,bot_*) of the PCB can be described individually in terms of the true strains on the top and bottom surfaces (ε*_t,top_* and ε*_t,bot_*) as
(2)$$\varepsilon_{a,top}=\varepsilon_{t,top}+\frac{\left(\gamma_{g}+\gamma_{w}\right)}{S_{g}}\Delta T$$
(3)$$\varepsilon_{a,bot}=\varepsilon_{t,bot}+\frac{\left(\gamma_{g}+\gamma_{w}\right)}{S_{g}}\Delta T$$

The strain data at various temperatures can be further converted into the bending strains (ε*_b_*) and bending curvatures (*k*) of the PCB by Equations (4) and (5), respectively.
(4)$$\varepsilon_{b}=(\varepsilon_{a,bot}-\varepsilon_{a,top})/2=(\varepsilon_{t,bot}-\varepsilon_{t,top})/2$$
(5)$$k=2\varepsilon_{b}/t=(\varepsilon_{t,bot}-\varepsilon_{t,top})/t$$
where *t* is the PCB thickness. Then, the out-of-plane displacement (deformation) *w* of the PCB under different temperatures can be further determined by Equation (6) with a given curvature *k* and distance *x* from the center.
(6)$$w=kx^{2}/2$$

As for the CTE measurement of the DIMM socket, the thermally induced strains across the thickness of the socket shown in Figure 2 consist of free thermal strain ε*_α_* from thermal expansion and pure bending strains (ε*_b_* and ε’*_b_*) from the curvatures in the x- and y-axis (*k_x_* and *k_y_*).

If the strain gauge is attached on the front surface in the neutral axis, the strain (ε*_G_*_3, _*_Front_*) obtained from such gauge can be written as
(7)$$\varepsilon_{G3,Front}=0+\varepsilon_{\alpha }+\varepsilon_{b}^{\prime }{}_{}$$

Similarly, the strain (ε*_G3, Backt_*) obtained from the back surface gauge is
(8)$$\varepsilon_{G3,Back}=0+\varepsilon_{\alpha }-\varepsilon_{b}^{\prime }{}_{}$$

The free thermal strain ε*_α_* can be determined by
(9)$$\varepsilon_{\alpha }=\frac{\varepsilon_{G3,Front}+\varepsilon_{G3,Back}}{2}$$

Thus, the CTE (α) for the socket is obtained as
(10)$$\alpha =\frac{\Delta \varepsilon_{\alpha }}{\Delta T}$$

If ordinary strain gauges (not more expensive thermal gauges) are used in the CTE measurement of materials (such as the PCB and DIMM socket in this study), the temperature effect on gauge grid metal material and connected lead wire should be taken into account according to Equation (1). The approach used here was to adopt the concept of a compensation CTE (α_comp_) by testing one known CTE material. A quartz (SiO_2_) plate with α = 0.5 ppm/°C was tested using front and back strain gauges, shown in Figure 3. The average strain is plotted against the temperature. The line slope of average strain is the apparent CTE of −5.72 ppm/°C, but the CTE of quartz is actually 0.5 ppm/°C. Therefore, the compensation CTE (α_comp_) obtained from our gauge system is 6.22 ppm/°C.

### 2.2. Shadow Moiré Measurements

Model PS600 provided by Akromatrix, one of the commercial shadow moiré systems, is shown in Figure 4a and was used in this study. The principle of the out-of-plane deformation measurement in the system is schematically shown in Figure 4b by illuminating the light through a mechanical grating on the surface of the specimen. The camera is for capturing the moiré fringe contours, while the movement of the stage driven by a servomotor is for phase shafting in order to enhance the spatial resolution of the measurement. This system features a spatial resolution of 2.5μm at s temperature range from 0 to 300 °C, with a heating rate of 2 °C/s and cooling rate of 1 °C/s. Two kinds of test samples, the bare PCB and a socket-PCB assembly with 6 sockets, shown in Figure 5a,b, respectively, were tested in the moiré system by measuring the surfaces of their PCBs.

### 2.3. Theoretical Analysis and FEM Simulation

For the theoretical analysis, a schematic of a quarter model of a socket-PCB assembly with one socket and its related geometric and material parameters are shown in Figure 6.

Based on the Suhir theory [13,14] and its modified version [15], the thermal deflection *w (x)* of the socket-PCB assembly can be revised by considering the beam-like deformation and expressed as
(11)$$w(x)=\frac{t\Delta \alpha \Delta T}{2\lambda D}\left[\frac{1}{2}x^{2}-\frac{\cosh k_{a}x-1}{k_{a}{}^{2}\cosh k_{a}l}\right]$$
where $t=t_{1}+t_{2}+t_{3}$; $k_{a}=\sqrt{\frac{\lambda }{\kappa }}$; $\kappa =\frac{t_{1}{}^{}}{3G_{1}b}+\frac{2t_{2}}{3G_{2}b}+\frac{t_{3}}{3G_{3}W}$; $G_{i}=\frac{E_{i}}{2\left(1+v_{i}\right)}$; $\lambda =\frac{1}{E_{1}t_{1}b}+\frac{1}{E_{3}t_{3}W}+\frac{t^{2}}{4D}$; and $D=\frac{E_{1}t_{1}^{3}b}{12}+\frac{E_{2}t_{2}^{3}b}{12}+\frac{E_{3}t_{3}^{3}W}{12}$.

The thermal load is
(12)$$\Delta T=T_{f}-T_{i}$$
where *T_f_* is a final temperature and *T_i_* is an initial one. The difference of thermal expansion coefficient (Δα) between the socket (α_1_) and the PCB (α_3_) is
(13)$$\Delta \alpha =\alpha_{3}-\alpha_{1}$$

The *E*, *ν*, *α*, and *t* denote the elastic modulus, Poisson’s ratio, the thermal expansion coefficient, and the thickness of each layer, respectively, with subscripts from 1 to 3 representing the socket, effective adhesive, and PCB. The *b* and *W* are the width of the socket (or effective adhesive) and the PCB, respectively. The above-mentioned equations can be used for thermal warpage calculations of the one-socket-PCB assembly.

For the FEM simulation, a quarter model for the one-socket-PCB assembly is shown in Figure 7a with two planes of symmetry and a zoomed-in area. The roomed-in area has two different geometries shown in Figure 7b: one with a full model (including socket housing, Cu pins, and solder joints) and the other with a simplified socket model (only including socket housing and effective material). Note that the simplified socket model used here is to prepare for analyzing more complicated structures of a multi-socket-PCB assembly. Material properties used in the theoretical analysis and FEM simulation are listed in Table 1, in which the α for the socket and PCB and the E for the socket were actually measured in this study, and the rest of them except for *E*_2_ were provided by material vendors. The *E*_2_ for the effective material of the pin solder joint is varied in the analyses and then approximated by the FEM simulation.

## 3. Results and Discussion

The obtained results of the measurements and modelling for the thermally induced warpages of the DIMM socket-PCB assembly will be extensively discussed individually in this section, including material property measurements of the socket and the PCB, FEM modelling, thermal warpage measurements, warpage comparisons from different analyses, and warpages of the socket-PCB assembly after solder reflow.

### 3.1. Material Property Measurements

(a)Elastic Modulus and CTE of DIMM Socket

Since the DIMM socket is a hollow structure with many Cu pins inside, it is difficult to calculate or measure its elastic modulus. The effective elastic modulus was adopted and determined by performing the three-point bending test for the socket shown in Figure 8a, associated with the related FEM modelling of the test in Figure 8b, and then the effective elastic modulus can be determined as 8.7 GPa from the load–displacement curves obtained from the test and FEM simulation shown in Figure 8c. Furthermore, both bending strain distributions shown in Figure 8d across the thickness of the socket with and without Cu pins from the FEM simulation reveal that the neutral axis (with zero strain) is located at half of the socket thickness. This helps the CTE measurement of the socket using the strain gauge. Since the data of the CTE are more influential than those of the elastic moduli in terms of the thermal warpage according to Equation (11), the CTEs of the socket and PCB should be precisely determined later.

Moreover, typical strain data from the front and back gauges located at the neutral axis of a socket during the second, third, and fourth thermal loading cycles were obtained and are shown in Figure 9a. Note that the average of the front and back strains represents the thermal expansion strain, and the slope of the curve is the CTE, based on Equations (9) and (10). The obtained CTEs need to be added to α_comp_ (=6.22 ppm/°C) to exclude the thermal effect on the gauge and the connected wires. Then, the CTEs of three sockets (no. 1 to no. 3), obtained from gauges for these three thermal cycles, are shown in Figure 9b with their average of 14 ppm/°C. However, the strain data from the front and back gauges for those three sockets during the first thermal loading cycle are shown in Figure 10a, which are drastically dissimilar to those in Figure 9a, especially for temperatures beyond 120 °C. The CTE data extracted from the first thermal cycle are further shown in Figure 10b in comparison with those from the second to fourth cycles. The CTEs of three sockets are found to be not stable for the first thermal cycle, especially at temperatures higher than 120 °C, but stable after the first thermal cycle. Such variability in the CTEs may be attributed to the partially cured polymer of the socket before the first thermal cycle.

(b)CTE of Bare PCB

The strain gauges were also applied for the CTE measurement of the bare PCB. Typical strain data from three pairs of the front and back gauges (located at G_A_, G_B_, and G_C_) on the PCB under thermal loading are shown in Figure 11a. The corresponding average strains are shown in Figure 11b for those three pairs of gauges during two thermal cycles and then the obtained average CTE of the bare PCB are about 18 ppm/°C. It is found that, unlike the socket, the CTE data of the bare PCB is much more stable and consistent during the thermal cycling.

### 3.2. FEM Modelling

(a)Model Validation and Comparison with Theory

To simplify the FEM calculation, the simplified socket model (shown in Figure 7b) was adopted in the simulation instead of the full socket model. The results of thermally induced warpages (at point a’) for a one-socket-PCB assembly are shown in Figure 12a with various effective moduli (E_2_) of the joint under different thermal loadings (ΔT). It is apparent that the warpage linearly depends on the thermal loading, but depends quite nonlinearly on the effective moduli of the joint. Meanwhile, for replacing the full socket model with the simplified one, the acceptable E_2_ should be determined and used for the FEM simulation afterward. For this, the thermal warpage obtained from the full socket model is also plotted in Figure 12a in comparison with those from the simplified one. It can be seen that the simplified socket model with E_2_ = 100 MPa can be representative of the full one with the same warpage values. Moreover, a further comparison of the out-of-plane deformations (along the line a-a’) of the one-socket-PCB assembly under thermal loading ΔT = 150 °C with E_2_ = 100 MPa is shown in Figure 12b from the FEM simulations with full and simplified socket models. The consistency of both models is ensured again. Therefore, it is feasible to apply the simplified socket model associated with E_2_ = 100 MPa to the rest of the FEM simulations in this study. The theoretical result of the deformation is also plotted in Figure 12b, compared with the FEM ones. It is seen that the theoretical solution, with slight overprediction, is consistent with both FEM results. For further parametric study, the effect of the width of the PCB on the thermal warpage of a one-socket-PCB assembly along the socket line is shown in Figure 13a from the theoretical and FEM analyses for E_2_ = 100 MPa and 1000 MPa. Both results indicate that the thermal warpage decreases with increasing the width of the PCB, and the discrepancy between both results is relatively lower for E_2_ = 1000 MPa. Furthermore, the effects of effective modulus E_2_ and thickness t_2_ are also demonstrated in Figure 13b from both analyses with W/b = 7.5 under ΔT = 150 °C. It can be seen that the thermal warpage decreases with increasing the thickness t_2_ but decreasing the effective modulus E_2_. However, both parametric effects become relatively insignificant as the effective modulus E_2_ is larger than 1000 MPa. It is noted from Figure 12 and Figure 13 that a sight overprediction of the theoretical solution in comparison with the FEM one, especially for a low effective modulus E_2_, is also found in the literature [16] due to the limitation of the plate or beam assumption used in the theoretical formulations. Although our concern in this study is mainly with the warpage of the assembly, interfacial stresses are also important and comprehensively discussed [17] in the joint of the assembly.

(b)Local Model vs. Full Model

Since the test specimen is cut from the full socket-PCB assembly, the local model representing the test specimen has to be ensured to have the same behavior as the full model. Both local and full models under thermal loading ΔT = 150 °C were analyzed by the FEM simulation. The results of the out-of-plane deformations along the socket line o-a and their contour maps for the socket-PCB assembly are shown in Figure 14. It is obvious that, even though both contour maps look quite different due to different settings of the zero-deformation positions, the out-of-plane deformations along the socket line for both models are almost identical. Therefore, investigating the test specimen by experimental and FEM analyses can help in understanding the thermal deformation behavior of the full socket-PCB assembly.

### 3.3. Thermal Warpage Measurement

(a)Bare PCB

The out-of-plane deformation contours of the bare PCB from the moiré measurement at various temperature are shown in Figure 15a, and the out-of-plane deformations of the bare PCB along three socket lines (A, B, and C) in which the sockets will be located are plotted in Figure 15b. Note that the positive value of the deformation is concave, while the negative value indicates convex. The key line is a line with six through vias for fixing the socket on the PCB before the solder reflow. It is shown that the bare PCB at room temperature (T = 30 °C) is slightly concave and becomes flatter after heating up. The thermally induced deformations of two bare PCB specimens along those three lines at various temperature loadings (ΔT) are shown in Figure 16a, which were extracted from the moiré data. The corresponding warpages of the two bare PCB specimens along the socket lines at various thermal loadings are shown in Figure 16b. The results indicate that the bare PCB subject to the thermal heat loading deforms convex with a socket line warpage value between −5 μm and −25 μm atΔT = 150 °C.

(b)Socket-PCB Assembly

The out-of-plane deformation contours of the six-socket-PCB assembly, obtained from moiré measurements at various temperatures, are shown Figure 17a and their out-of-plane deformations of this assembly along the socket lines (A to C) are also plotted in Figure 17b with setting zero deformation at the key line. It can be seen that the concave-shape deformation increases with an increase in temperature. Furthermore, by subtracting the deformation at 30 °C, thermally induced deformations for two samples of the six-socket-PCB assembly along the socket lines at various temperature loadings (ΔT), obtained from moiré measurements, are shown in Figure 18a, and the corresponding warpages of those two samples along the socket lines at various temperature loadings (ΔT) are presented in Figure 18b. Those results indicate that the thermally induced warpage along the socket lines increases linearly with temperature loading and its variation is less significant between the different socket lines as well as between the samples.

### 3.4. Comparisons of Thermal Warpages from Moiré, Gauges, Theory, and FEM

The thermally induced deformations of the six-socket-PCB assembly along the socket lines (line o-a) at various temperature loadings (ΔT) are shown in Figure 19a from moiré measurements, theory, and FEM simulation, and in Figure 19b for their corresponding warpages along the socket lines at various temperature loadings (ΔT). Note that moiré data presented here have been modified by considering the thermal warpage of the bare PCB observed in Figure 16b, while the *b* = 6 × 6.4 mm = 38.4 mm and *W* = 105 mm were used in the theoretical analysis. It is found that all moiré, theoretical, and FEM results are reasonably consistent, either for thermally induced deformations or warpage along the socket lines. Moreover, the front and back gauges located at the PCB near the middle of the socket lines on the socket-PCB assembly were used for measuring the bending strains (or curvature) under two thermal loading cycles. The obtained data are shown in Figure 20a for the apparent strains from the front and back gauges under two thermal cycles. Additionally, the gauge-determined corresponding warpage along the socket line is obtained and shown in Figure 20b by converting those strains to bending strain, curvature, and warpage data using Equations (4)–(6), respectively. It is found that the thermally induced warpage along the socket line, measured by the gauges, increases linearly with temperature at the beginning, but starts dropping near T= 180 °C. This turning-around mechanism cannot be clearly understood at this moment but could be discussed later. It is also shown that the gauge results are in good agreement with the moiré data, especially for those along the socket line A.

### 3.5. Warpage of Socket-PCB Assembly after Solder Reflow

The cooling-rate effect on thermally induced warpages of the six-socket-PCB assembly after solder reflow was also studied. The gauge results of the warpage along the socket lines with two cooling cycles each at two different cooling rates (0.6 °C/min and 1.2 °C/min, corresponding, respectively, to 6 h and 3 h of cooling time from 250 °C to 30 °C) are shown in Figure 21, compared with those from the theory and FEM simulation. Note that the negative value of the warpage indicates the concave-down deformation. It is found that the zero thermal warpages are maintained down to 180 °C from 250 °C during the cooling, and those warpages have lower values with lower cooling rates, but the theoretical and FEM values give an upper bound for those. The reason behind it is that the lead-free solder (with a melting temperature of 217 °C) used here is soft with a strong creep (visco-plastic) behavior at high temperatures and it could not generate a constraint between the socket and PCB until the temperature reached 180 °C. Such a behavior has also been observed in a PBGA package-PCB assembly [11]. Additionally, the cooling-rate dependency of the thermal warpage is caused by the creep behavior of the solder joint. However, since the fast cooling rate of around 60 °C/min is generally adopted in the reflow process, the FEM simulation without considering the creep behavior might give a good prediction of the thermal warpage of the socket-PCB assembly after the solder reflow. From the present observation of the zero warpage induced beyond 180 °C, it might indicate there is no constraint between the socket and PCB due to the very soft material property and high creep rate of the solder joints. That could be a reason for the warpage drop in the strain gauge data beyond 180 °C, as observed in Figure 20b.

Based on the zero warpage occurring at the temperature of 180 °C, the effects of the thickness of the PCB (t_PCB_) and mismatch of CTE (Δα) between the socket and PCB on the thermally induced warpages of the PCB specimen with six sockets along the socket line (line o-a) and in the full field (in inserted graph) are shown in Figure 22 from the FEM simulation for thermal loading ΔT= 30 °C−180 °C = −150 °C. The warpage data can represent those of the specimen cooling down after the reflow process. The thicknesses of the PCB here are taken based on the possible values in the application. The results indicate that the thermal warpage is linearly dependent on the Δα, but nonlinearly dependent on the t_PCB_. The linear dependence on the Δα and the nonlinear dependence on t_PCB_ can also be observed in Equation (11) for the thermal deflection from the theoretical solution. Moreover, the effects of both parameters are also applied to the specimen with 24 sockets on the PCB (like the configuration in Figure 1b). The results of the thermal warpage at the corner of the PCB (or the maximum warpage) and the full-field deformation are shown in Figure 23 with various t_PCB_ and Δα from the FEM simulation after solder reflow. It can be seen that, after the solder reflow, the socket-PCB assembly deforms cylindrically with a maximum warpage at the corner of the PCB and the maximum thermal warpage also depends linearly on the Δα and nonlinearly on the t_PCB_. It is specially noted that the FEM results shown in Figure 22 and Figure 23 were obtained by assuming that the bare PCB is always flat during the solder reflow heating and cooling process.

## 4. Conclusions

The problem of thermal warpage for the DIMM socket-PCB assembly after the solder reflow processes has been extensively investigated experimentally, theoretically, and numerically, especially along the socket lines and over the entire assembly. In this study, a beam-like theoretical solution to this problem has been newly proposed by modifying the existing Suhir theory. The theoretical solution validated by the finite element method (FEM) simulation has provided the detailed mechanics, including the critical parameters, such as the coefficient of thermal expansion (CTE, α), elastic modulus (E_2_), thickness (t_2_) of the effective material (which is a combined material of a solder joint and Cu pin), and width of the PCB. The CTEs of the socket and the bare PCB have been obtained as 14 and 18 ppm/°C by the strain gauge measurement, respectively. However, inconsistent data for the CTE were found for the sockets in the measurement of the first thermal cycle, but the CTE became constant after the first thermal cycle. Meanwhile, the effective elastic moduli of the socket and the combined material of solder joint and Cu pin have been determined by a hybrid method (combined with the experiment and FEM) and FEM simulations with a simplified model, respectively. Furthermore, the cylindrical-like thermal deformation and warpage of the bare and socket-PCB have been measured by the moiré experiment, and the thermally induced warpages of the socket-PCB assembly are also consistent with those from the theory and FEM simulations. Moreover, the result from the moiré-consistency strain gauge measurement for the thermal warpage of the socket-PCB assembly has revealed that the thermal warpage starts at 180 °C during the process of cooling down from the reflow peak temperature (250 °C) and is dependent on the cooling rate due to the nature of the high creep (visco-plastic) behavior of the solder material. Finally, the thermally induced warpages of the full socket-PCB assemblies along the socket lines and over the entire assembly after the solder reflow process have been provided in this study by the validated FEM simulation in terms of the PCB thickness (t_PCB_) and the CTE mismatch (Δα) between the socket and PCB for future design and verification.

## Figures and Tables

**Figure 1 materials-16-03233-f001:**
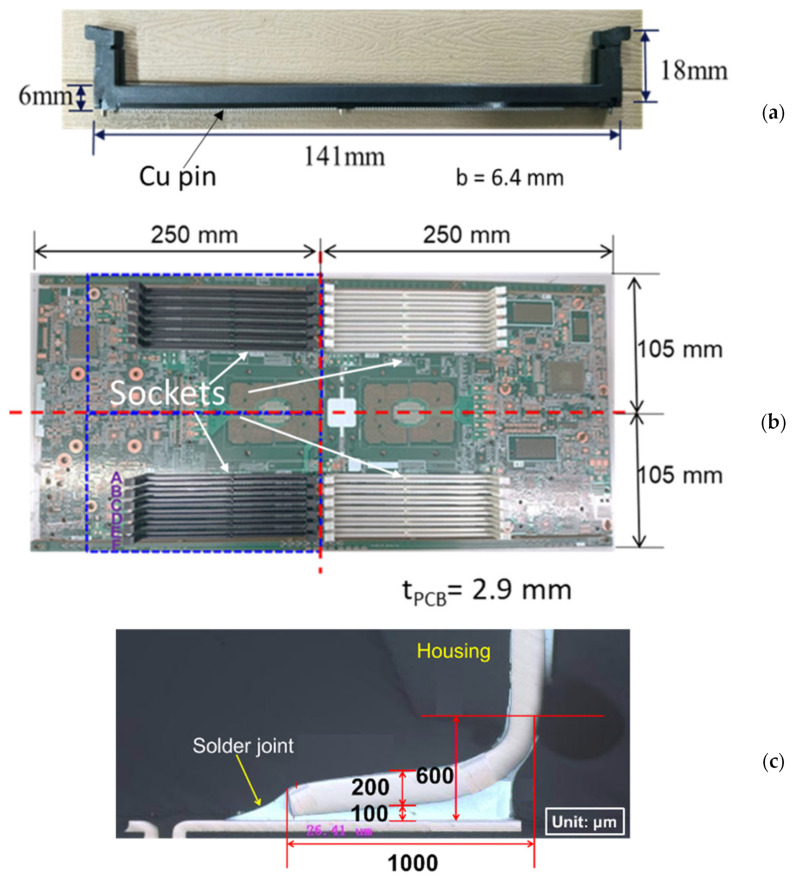
Configurations of (**a**) a DIMM socket with 288 Cu pins and width of 6.4 mm, (**b**) a socket-PCB assembly with 24 sockets (a full assembly), and (**c**) a detailed solder joint of a Cu pin for the DIMM socket on the PCB.

**Figure 2 materials-16-03233-f002:**
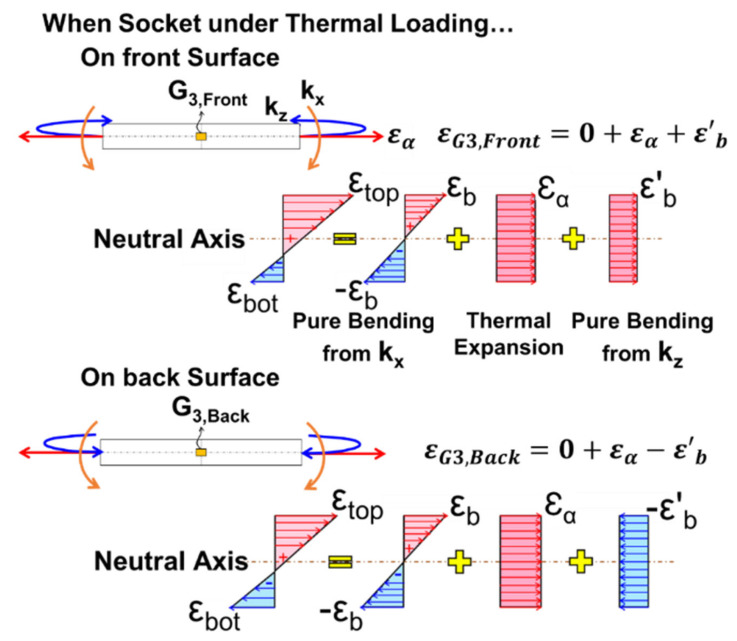
Strain distributions across the thickness of the DIMM socket under thermal loading.

**Figure 3 materials-16-03233-f003:**
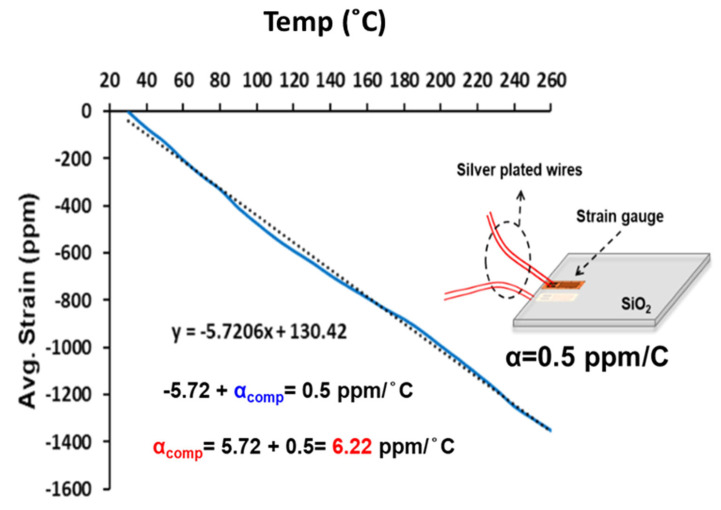
Calibration of strain gauge measurement for CTE measurement.

**Figure 4 materials-16-03233-f004:**
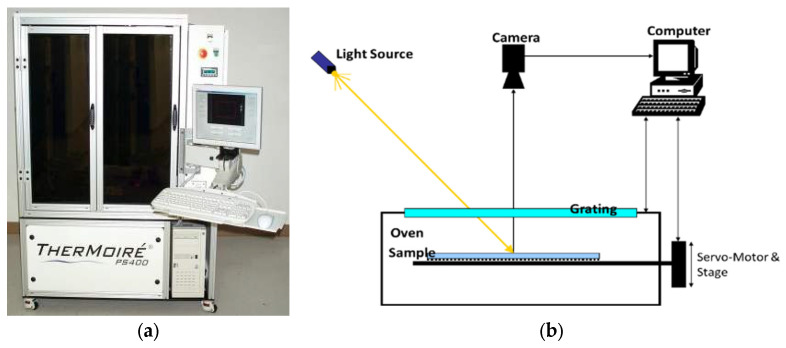
(**a**) Commercial shadow moiré system, and (**b**) schematic of the principle of out-of-plane deformation measurement in the system.

**Figure 5 materials-16-03233-f005:**
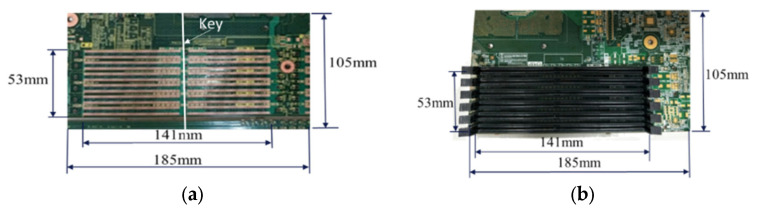
Configurations of the test specimens: (**a**) the bare PCB and (**b**) a socket-PCB assembly with 6 sockets (a local assembly).

**Figure 6 materials-16-03233-f006:**
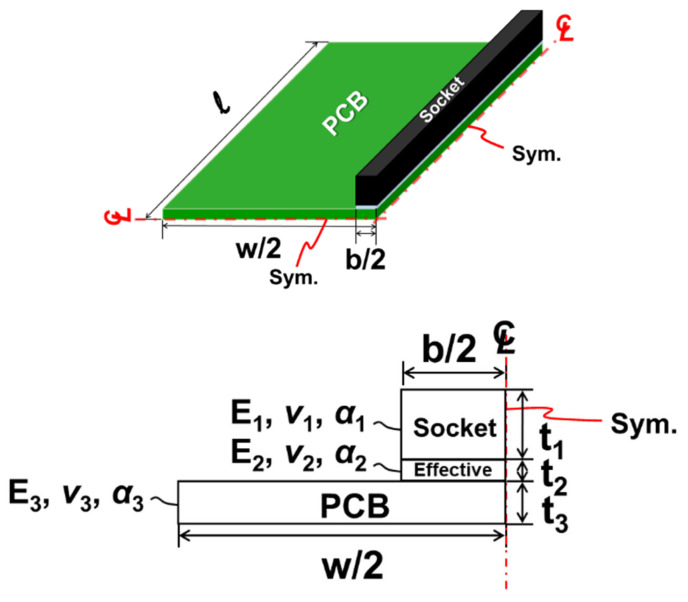
Quarter model of a socket-PCB assembly with one socket and its related geometric and material parameters in the theoretical formulation.

**Figure 7 materials-16-03233-f007:**
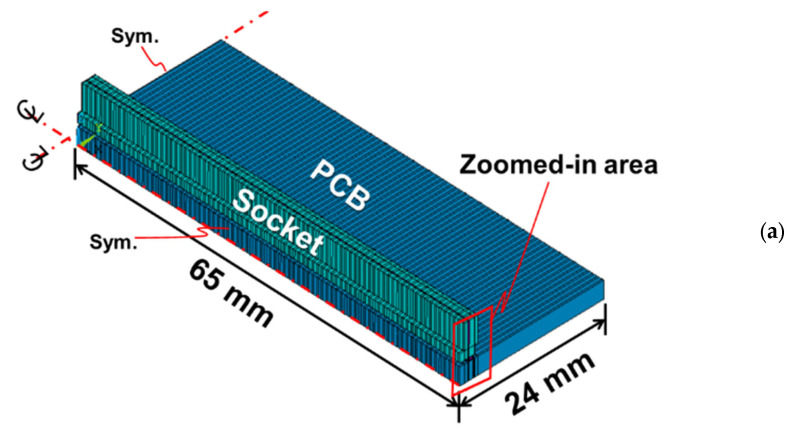
(**a**) A quarter FEM model for a one-socket-PCB assembly (**b**) with full and simplified socket models in the zoomed-in area.

**Figure 8 materials-16-03233-f008:**
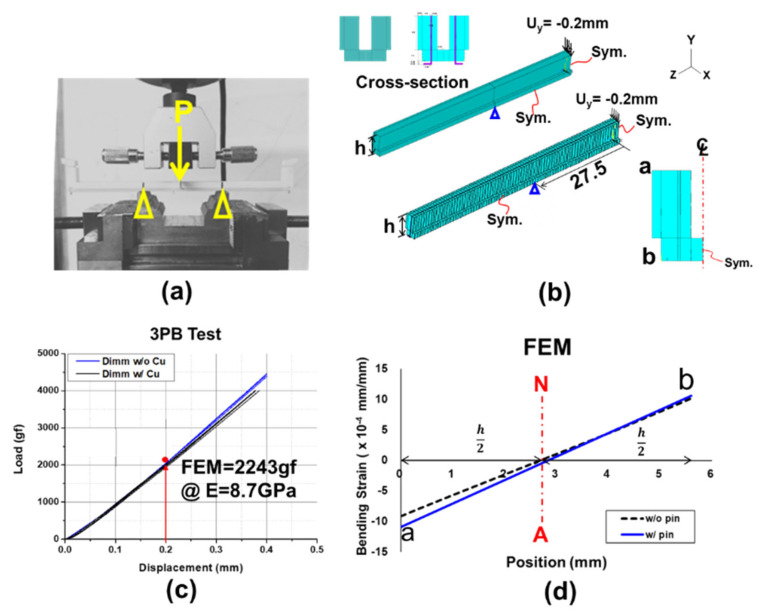
(**a**) Three-point bending test for a socket, (**b**) FEM modelling of the test, (**c**) determination of effective elastic modulus from load–displacement curves obtained by the test and FEM simulation, and (**d**) bending strain distributions across the thickness of the socket with and without Cu pins from the FEM simulation.

**Figure 9 materials-16-03233-f009:**
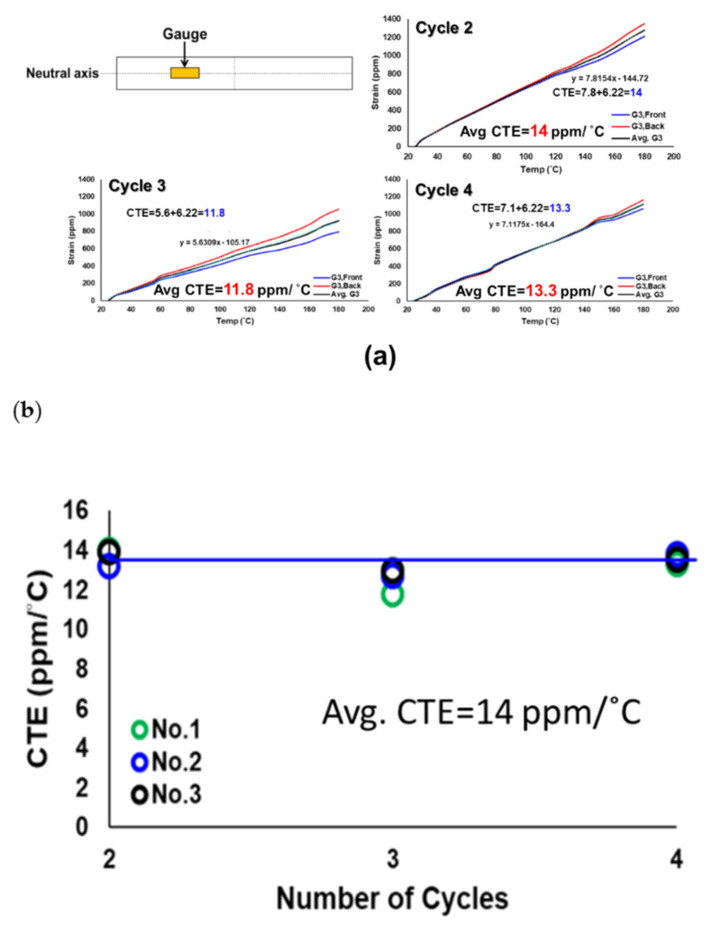
(**a**) Typical strain data from the front and back gauges located at the neutral axis of a socket during the second, third, and fourth thermal cycling loading, and (**b**) the CTEs of three sockets, obtained from gauges for these three thermal cycles.

**Figure 10 materials-16-03233-f010:**
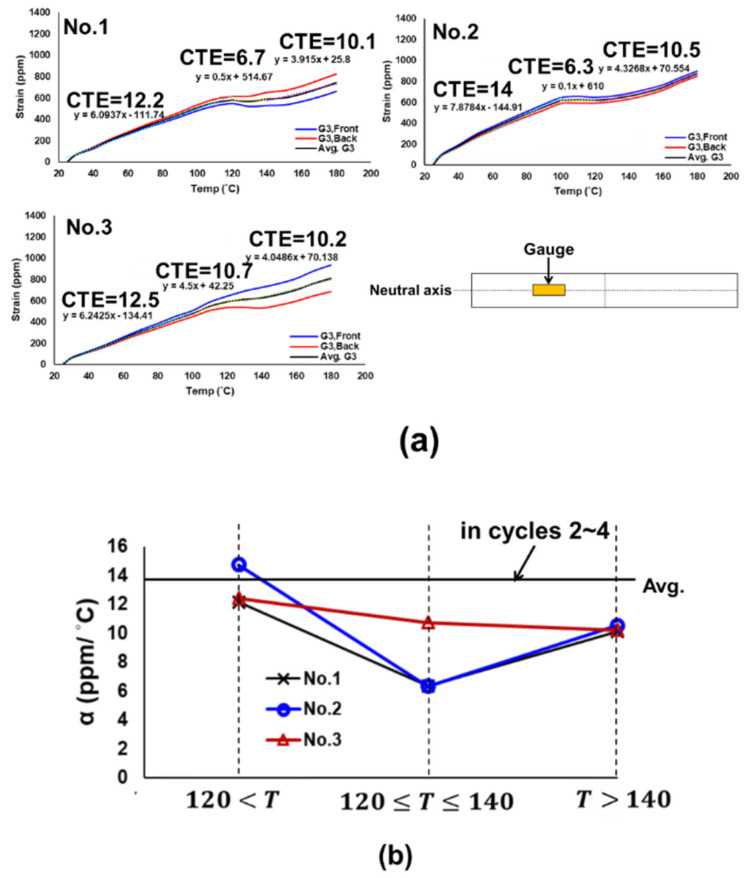
(**a**) Strain data from the front and back gauges for three sockets during the first thermal loading cycle, and (**b**) temperature-dependent CTEs of three sockets obtained from gauges for the first thermal cycle and their comparison with those for the second to fourth cycles.

**Figure 11 materials-16-03233-f011:**
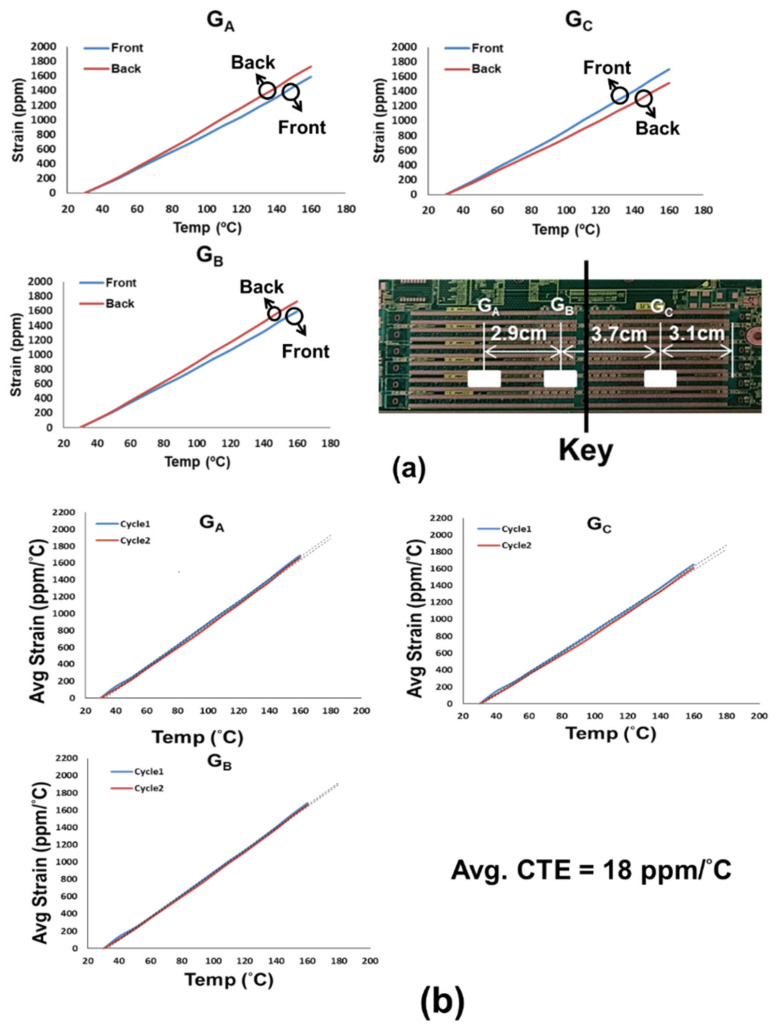
(**a**) Typical strain data from three pairs of front and back gauges (G_A_, G_B_, and G_C_) on the bare PCB under thermal loading, and (**b**) average strains from front and back gauges for those three pairs during two thermal cycles and the CTEs of the PCB obtained from those average strain data.

**Figure 12 materials-16-03233-f012:**
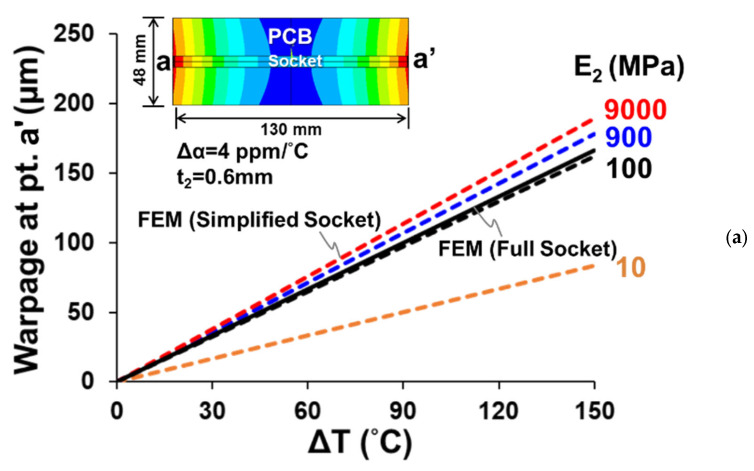
(**a**) Thermally induced warpage (plus an inset of full-field deformation contours) of a one-socket-PCB assembly with various effective moduli of the joint under different thermal loadings (ΔT) from FEM with full and simplified socket models, and (**b**) the out-of-plane deformations (along the line a-a’) of the one-socket-PCB assembly under thermal loading ΔT = 150 °C, from the theory and FEM simulations with full and simplified socket models.

**Figure 13 materials-16-03233-f013:**
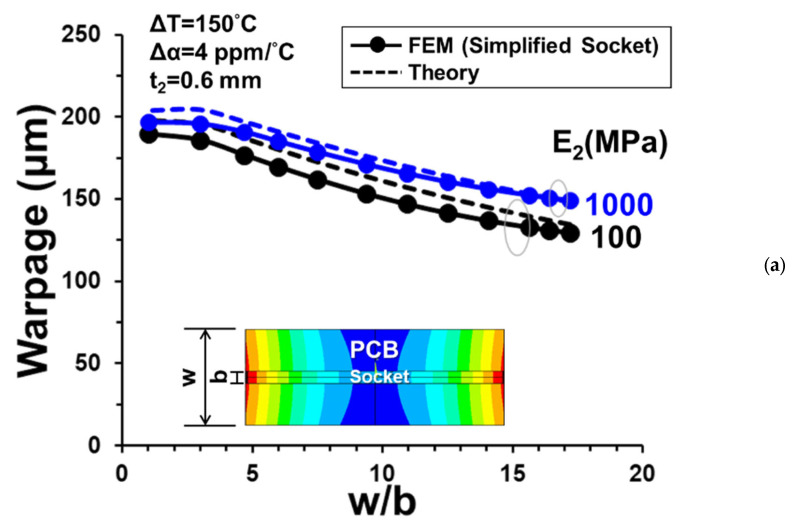
Thermally induced warpage of a one-socket-PCB assembly (**a**) with various widths of the PCB for E_2_ = 100 MPa and 1000 MPa, and (**b**) with various E_2_ and t_2_ but W/b = 7.5 under ΔT = 150 °C from the FEM with simplified socket model and the theory.

**Figure 14 materials-16-03233-f014:**
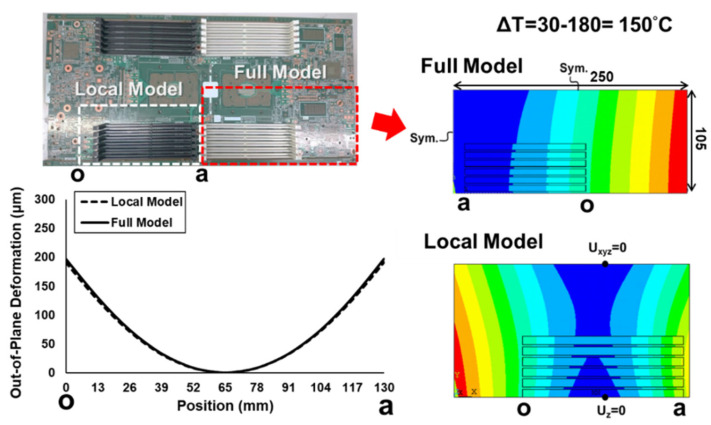
The out-of-plane deformations along the socket line o-a and their contour maps for the socket-PCB assembly under thermal loading ΔT = 150 °C, determined from the FEM simulation using a local model and a full model.

**Figure 15 materials-16-03233-f015:**
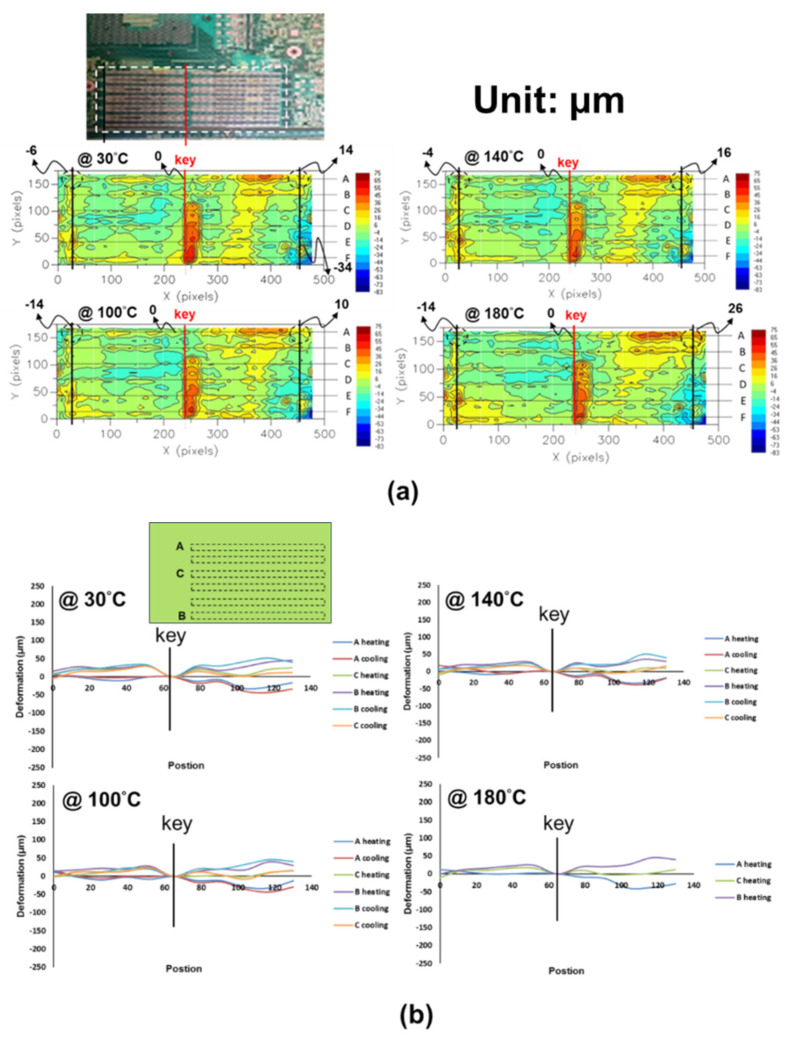
(**a**) Out-of-plane deformation contours (in dotted line box) for the bare PCB at various temperatures, obtained from moiré measurements, and (**b**) out-of-plane deformations of the bare PCB along three lines (A, B, and C) in which the sockets will be located.

**Figure 16 materials-16-03233-f016:**
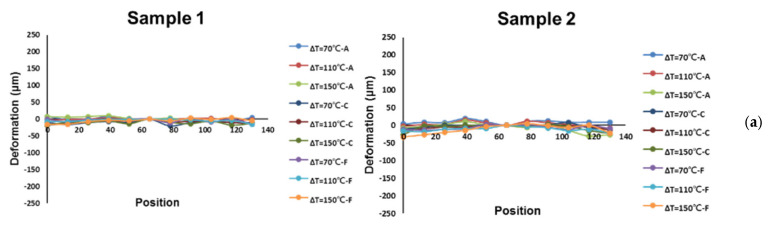
(**a**) Thermally induced deformations of two bare PCB specimens along the socket lines at various temperature loadings, obtained from moiré measurements, and (**b**) the corresponding warpages of the two bare PCB specimens along the socket lines (A, B, and C) at various temperature loadings.

**Figure 17 materials-16-03233-f017:**
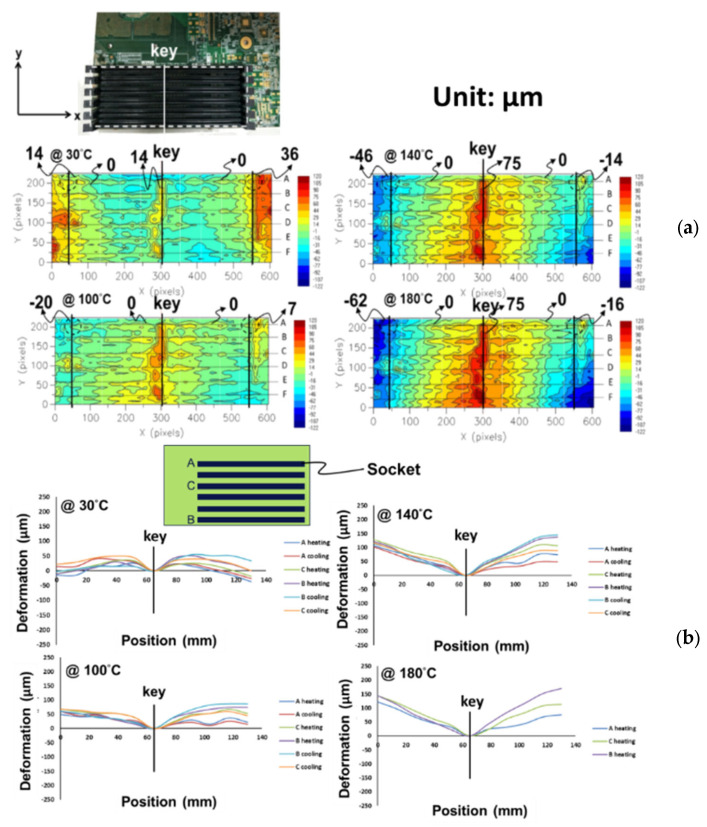
(**a**) Out-of-plane deformation contours (in dotted line box) for the six-socket-PCB assembly at various temperatures, obtained from moiré measurements, and (**b**) out-of-plane deformations of the socket-PCB assembly along the lines in which the sockets are located.

**Figure 18 materials-16-03233-f018:**
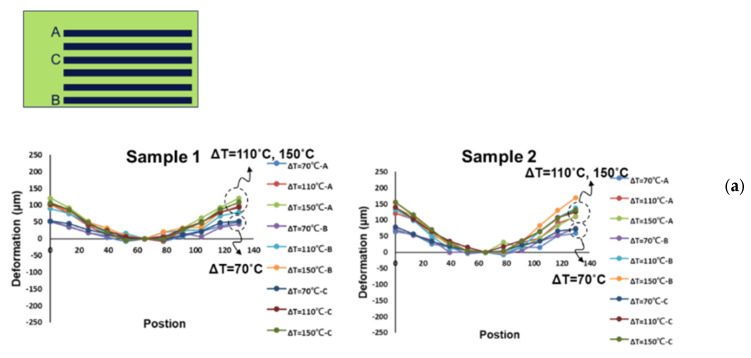
(**a**) Thermally induced deformations for two samples of the six-socket-PCB assembly along the socket lines at various temperature loadings (ΔT), obtained from moiré measurements, and (**b**) the corresponding warpages of the two samples along the socket lines at various temperature loadings (ΔT).

**Figure 19 materials-16-03233-f019:**
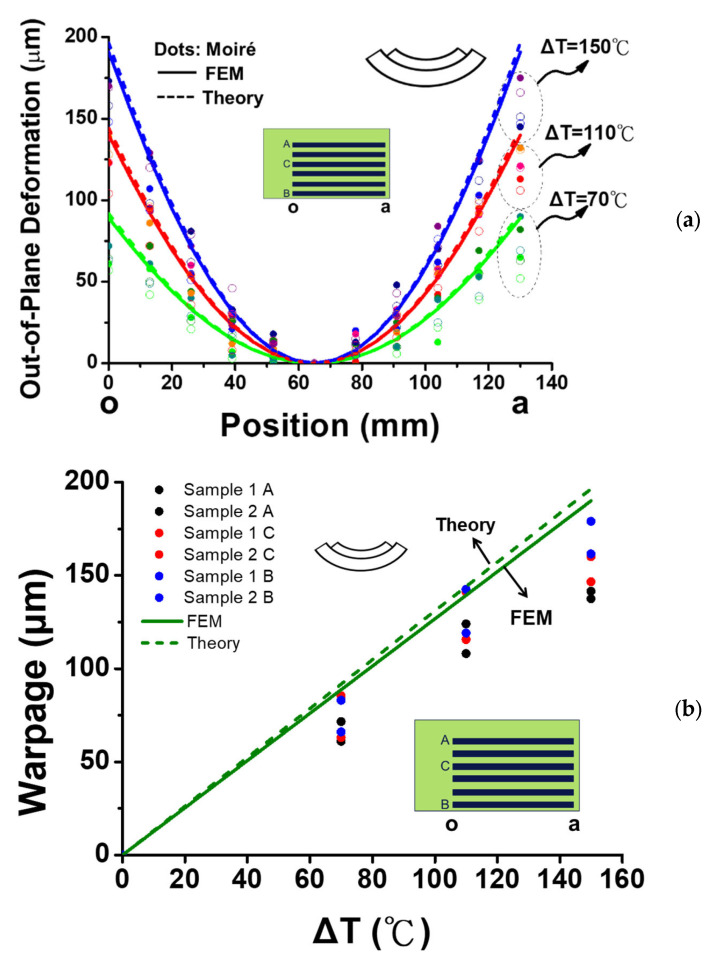
(**a**) Comparison of thermally induced deformations of the six-socket-PCB assembly along the socket lines (line o-a) at various temperature loadings (ΔT), obtained from moiré measurements, theory, and FEM simulations, and (**b**) the corresponding warpage comparison of the six-socket-PCB assembly along the socket lines at various temperature loadings (ΔT).

**Figure 20 materials-16-03233-f020:**
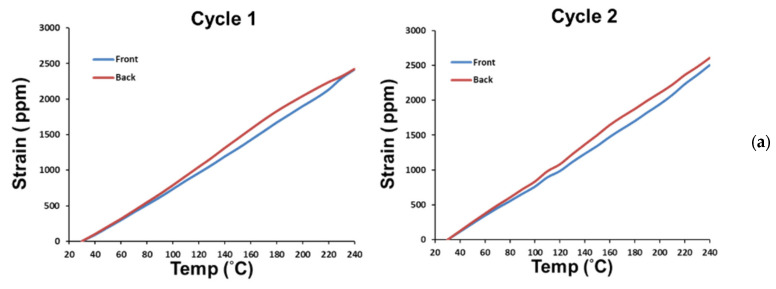
(**a**) Strain data of the front and back gauges on the six-socket-PCB assembly under two thermal loading cycles, and (**b**) the gauge-determined corresponding warpages of the six-socket-PCB assembly along the socket lines, compared with moiré results.

**Figure 21 materials-16-03233-f021:**
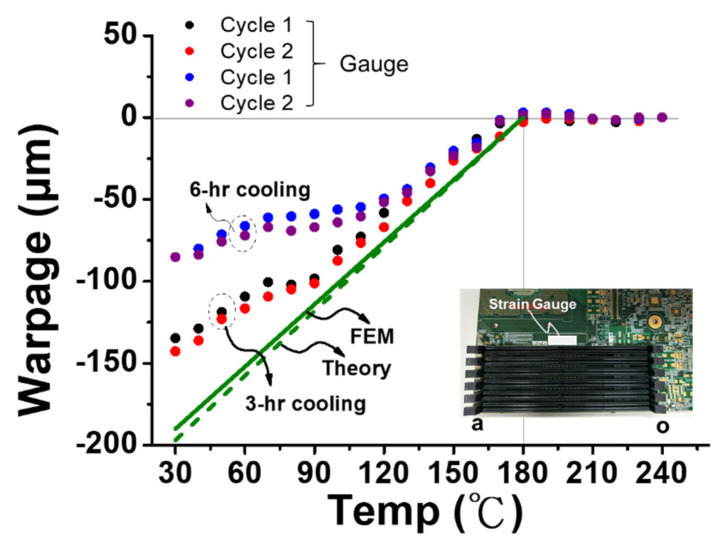
The cooling rate effect on thermally induced warpages of the six-socket-PCB assembly along the socket lines after solder reflow, from strain gauge, theory, and FEM simulation.

**Figure 22 materials-16-03233-f022:**
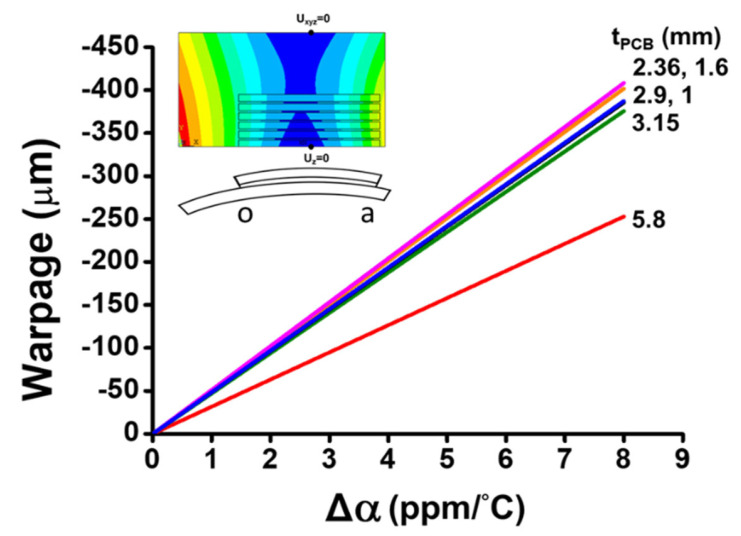
The effects of the thickness of the PCB (t_PCB_) and mismatch of CTE (Δα) on thermally induced warpages of the PCB specimen with six sockets along the socket lines (line o-a) from the FEM simulation for ΔT= 30 °C −180 °C = −150 °C.

**Figure 23 materials-16-03233-f023:**
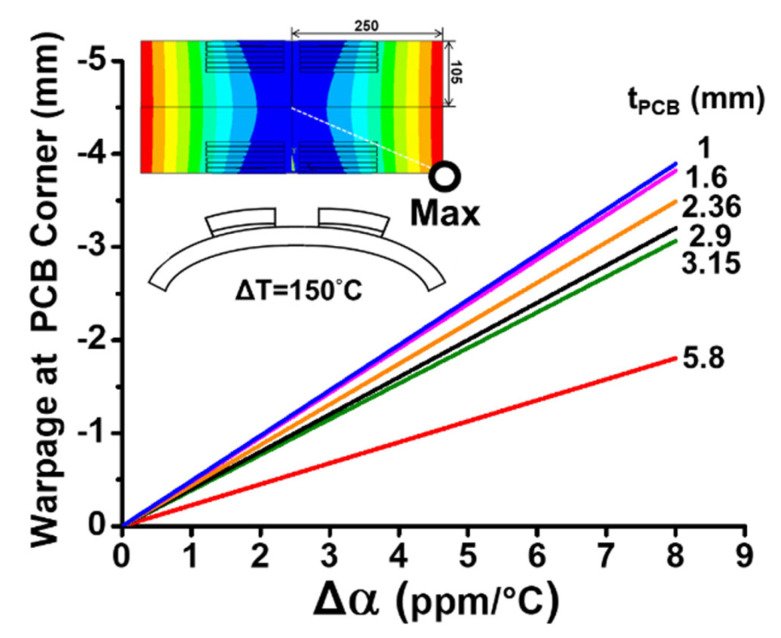
The effects of the thickness of the PCB (t_PCB_) and mismatch of CTE (Δα) on thermally induced warpages at the corner of the PCB for the socket-PCB assembly with 24 sockets from the FEM simulation ΔT = 30 °C −180 °C = −150 °C.

**Table 1 materials-16-03233-t001:** Material properties for the theoretical analysis and FEM simulation. (Note: The α for the socket and PCB and the E for the socket were measured in this study, while the rest of them except for E_2_ were provided by material vendors.)

Material	E (GPa)	α (ppm/°C)	*ν*
Socket	8.7	14	0.3
Pin Solder Joint (Effective)	E_2_	8.1	0.3
PCB	20	18	0.42
Cu Pin	120	16.9	0.3
Solder	30	22	0.3

## Data Availability

On inquiry, the data presented in this study is available from the authors.

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
