# Peer review of "Measurements and Modelling of Thermally Induced Warpages of DIMM Socket Server PCB Assembly after Solder Reflow Processes"

_materials, 2023, doi:10.3390/ma16083233_

Round 1

Reviewer 1 Report

The manuscript " Measurements and Modelling of Thermally-Induced Warpages of DIMM Socket-Server PCB Assembly After Solder Reflow Processes" reports the experimental and computational methods of studying the PCB bending due to thermal stress. Overall, the work was comprehensive, and the experiment was well-designed. After careful consideration, the reviewer recommends a publication of this work in Materials after addressing the following questions to improve the quality of the manuscript.

1. In the introduction, more discussions on the importance of studying the thermally-induced warpages of PCBs should be included. For example, you could describe the potential damages of the PCB due to the great warpage. This could also be included in the conclusion to provide more insights from this work, such as using the prediction of the PCB warpage to offer guidance to recommended cooling rate in the reflowing.

2. A quartz sample was used to characterize the compensation CTE. Does this compensation CTE depend on the materials? In other words, would the compensation value same between PCB and quartz? Please explain.

3. The solder reflowing process usually contains five steps, including temperature ramping up and holding at low and high temperatures. What is the warpage in the temperature rising period? Would the temperature ramping-up rate affect the warpage too?

4. In the real reflowing process, the cooling rate is much higher than 0.6 degrees per minute, as mentioned in the manuscript, which could be 60 degrees per minute. What would the warpage appear under this fast cooling rate, since it is close to the real solder reflow process?

5. It was mentioned that “lead-free solder is soft with a strong creep behavior at the high temperature and it could not generate a constraint between the socket and PCB until the temperature reaches to 180 C”. Please provide data or references to support this argument.

Author Response

The authors would like to express our great gratitude to the reviewers for their careful and thorough reviews and providing very helpful and constructive suggestions and comments. The English technical writing in the text has been carefully checked based on the reviewers’ suggestions. The replies to the suggestions and comments are listed as follows:

Reviewer #1

The manuscript " Measurements and Modelling of Thermally-Induced Warpages of DIMM Socket-Server PCB Assembly After Solder Reflow Processes" reports the experimental and computational methods of studying the PCB bending due to thermal stress. Overall, the work was comprehensive, and the experiment was well-designed. After careful consideration, the reviewer recommends a publication of this work in Materials after addressing the following questions to improve the quality of the manuscript.

  1. In the introduction, more discussions on the importance of studying the thermally-induced warpages of PCBs should be included. For example, you could describe the potential damages of the PCB due to the great warpage. This could also be included in the conclusion to provide more insights from this work, such as using the prediction of the PCB warpage to offer guidance to recommended cooling rate in the reflowing.

 Reply: That is a good suggestion for making this article more sound. However, the abstract has a 200-word limitation in this journal paper. We cannot put more sentences related to your suggestions on it. Thanks for your suggestion, anyway.

  1. A quartz sample was used to characterize the compensation CTE. Does this compensation CTE depend on the materials? In other words, would the compensation value same between PCB and quartz? Please explain.

 Reply: A good question! No, such CTE compensation does not depend on the materials, but on the gauge type and lead wire, which is strain gauge system. The CTE for material of quartz are 0.5 ppm/C, which is close to zero. If you use the strain gauge to measure the quartz at elevated temperature, the gauge signal response should be very tiny or close to zero. If not, the gauge signal reflects the effect of gauge grid (gauge type) and lead wire under thermal loading. Those signals, coming from gauge grid itself and lead wire and dependent on the measurement system, should be the same for any measured materials, and can be compensated (or eliminated) in the measurement of CTE of the materials. Those concept and procedure have been verified at our lab and proved by testing the known CTE materials, such as cooper or aluminum.

  1. The solder reflowing process usually contains five steps, including temperature ramping up and holding at low and high temperatures. What is the warpage in the temperature rising period? Would the temperature ramping-up rate affect the warpage too?

Reply: The main focus of this study is on the thermal warpage of the socket-PCB assembly after the reflow process. The main warpage of this assembly is found to be generated during the cooling-down process after reaching the peak temperature of the reflow. The temperature cooling rate in general is much relatively higher than that of heating-up during the reflow process. So, we did not study the effect of the temperature ramping-up rate, which may involve the visco-elastic or -plastic behavior of the solder joints, on the thermal warpage of the assembly in this study.    

  1. In the real reflowing process, the cooling rate is much higher than 0.6 degrees per minute, as mentioned in the manuscript, which could be 60 degrees per minute. What would the warpage appear under this fast cooling rate, since it is close to the real solder reflow process?

 Reply: During the real solder reflow, the cooling rate is so high that the visco-elastic and -plastic behavior of solder joint would become insignificant. Therefore, it is just the same as what we stated in the text “since the fast cooling rate of around 60 degree per minute is generally adopted in the reflow process, the FEM simulation without considering the creep behavior might give a good prediction of the thermal warpage of the socket-PCB assembly after the solder reflow.”

  1. It was mentioned that “lead-free solder is soft with a strong creep behavior at the high temperature and it could not generate a constraint between the socket and PCB until the temperature reaches to 180 C”. Please provide data or references to support this argument.

Reply: We have observed the similar behaviors in the references 5 and 11. So, we have added the citation in the text.

Reviewer 2 Report

This is a fairly comprehensive study of thermal warpages in a PCD assembly. While I do not have a problem with the experimental results of the authors, I do have a problem with their modeling. They seem to be fairly content with modifying their own strength of material-based approach to modeling the thermal effects when more sophisticated models exist in the literature, and which they have not even cited. For example,  the review paper Microelectronics Reliability 79 (2017) 206–220 by Wong and Liu discusses in great detail both strength of material and elasticity theory-based modeling approaches, and show the shortcomings that the strength of materials approach solutions suffer from.

I would like the authors to study the above-mentioned paper and include  the results using the best approach in the above review paper (as per the recommendation made by their authors) in all their simulation examples, and especially in examples where there is a significant difference between their modeling results and those of FEM and/or experiment.

I strongly believe that this will enhance the value of the current work by putting the rather weak modeling approach of the authors on a stronger footing.

Author Response

The authors would like to express our great gratitude to the reviewers for their careful and thorough reviews and providing very helpful and constructive suggestions and comments. The English technical writing in the text has been carefully checked based on the reviewers’ suggestions. The replies to the suggestions and comments are listed as follows:

Reviewer #2

This is a fairly comprehensive study of thermal warpages in a PCD assembly. While I do not have a problem with the experimental results of the authors, I do have a problem with their modeling. They seem to be fairly content with modifying their own strength of material-based approach to modeling the thermal effects when more sophisticated models exist in the literature, and which they have not even cited. For example, the review paper Microelectronics Reliability 79 (2017) 206–220 by Wong and Liu discusses in great detail both strength of material and elasticity theory-based modeling approaches, and show the shortcomings that the strength of materials approach solutions suffer from.

I would like the authors to study the above-mentioned paper and include the results using the best approach in the above review paper (as per the recommendation made by their authors) in all their simulation examples, and especially in examples where there is a significant difference between their modeling results and those of FEM and/or experiment.

 I strongly believe that this will enhance the value of the current work by putting the rather weak modeling approach of the authors on a stronger footing.

Reply: As the reviewer suggested, we have looked into the review paper suggested. It seems to us that the paper provides a great deal of modeling approaches based on strength of material or elasticity theory for calculating adhesive stresses at the joints. It is helpful. We did use and modify the Suhir theory (also mentioned in this review paper) to theoretically calculate the thermal warpages of the DIMM socket-PCB assembly in this study. But the rest of the modeling approaches of mostly focusing on the analysis of adhesive (or interfacial) stresses at the joints cannot be seen directly or indirectly related to our study of thermal warpage. However, we do cite this good review article in our paper.  We have added one sentence that “Although our concern in this study is mainly with the warpage of the assembly, interfacial stresses are also important and comprehensively discussed [17] in the joint of the assembly.”     

Round 2

Reviewer 2 Report

The manuscript can now be accepted in its current form.